# Combining high-resolution scanning tunnelling microscopy and first-principles simulations to identify halogen bonding

James Lawrence [1], Gabriele C. Sosso [1,2✉], Luka Đorđević [3], Harry Pinfold[1], Davide Bonifazi [3✉] & Giovanni Costantini [1✉]

Scanning tunnelling microscopy (STM) is commonly used to identify on-surface molecular self-assembled structures. However, its limited ability to reveal only the overall shape of molecules and their relative positions is not always enough to fully solve a supramolecular structure. Here, we analyse the assembly of a brominated polycyclic aromatic molecule on Au(111) and demonstrate that standard STM measurements cannot conclusively establish the nature of the intermolecular interactions. By performing high-resolution STM with a CO-functionalised tip, we clearly identify the location of rings and halogen atoms, determining that halogen bonding governs the assemblies. This is supported by density functional theory calculations that predict a stronger interaction energy for halogen rather than hydrogen bonding and by an electron density topology analysis that identifies characteristic features of halogen bonding. A similar approach should be able to solve many complex 2D supramolecular structures, and we predict its increasing use in molecular nanoscience at surfaces.

[1] Department of Chemistry, University of Warwick, Gibbet Hill Road, Coventry CV4 7AL, UK. [2] Centre for Scientific Computing, University of Warwick, Gibbet Hill Road, Coventry CV4 7AL, UK. [3] School of Chemistry, Cardiff University, Park Place Main Building, Cardiff CF10 3AT, UK. ✉email: G.Sosso@warwick.ac.uk; BonifaziD@cardiff.ac.uk; G.Costantini@warwick.ac.uk

In order to engineer robust functional molecular nanostructures at surfaces, self-assembly via strong, directional intermolecular forces is often required. Both halogen[1] and hydrogen bonding[2] possess these characteristics to different extents, offering different potential nanofabrication pathways and allowing for a diverse set of molecular moieties that can participate in the formation of supramolecular architectures. There are many examples in the literature of two-dimensional (2D) hydrogen bonded (HB) assemblies on surfaces that have been examined with surface science techniques[3–6]. Halogen-bonded (XB) molecular nanostructures[1,7–9] are considerably less common in on-surface 2D systems but are increasingly being studied and seen as an important addition to the 'toolbox' of supramolecular assembly. Typically, iodine[10–14] and bromine atoms[15–19] are used as halogen bond donors due to their strong polarisability, but examples of chlorine[20] and fluorine[21,22] based assemblies have also been reported. Halogen bonds can also involve halogen atoms acting simultaneously as both donor and acceptor[11,16,17,23,24], with the positive electrostatic potential of the sigma hole oriented towards the central 'belt' of negative electrostatic potential found on an adjacent halogen atom. Alternatively, nitrogen[10,12,13,19] and oxygen[17,25–27] atoms act as acceptor sites in heteromolecular assemblies.

2D assemblies are commonly characterised on surfaces with scanning probe microscopy (SPM) techniques such as scanning tunnelling microscopy (STM) and atomic force microscopy (AFM). In recent years, higher resolution forms of SPM have become available[28–32] that make use of functionalised tips (such as CO, Xe, $D_2$, $H_2$, Br, and CuO)[28,30,31,33,34] to reveal the internal structure of molecules adsorbed on surfaces with astounding clarity[29]. Non-contact AFM (NC-AFM) and high-resolution STM (HR-STM) have been used to resolve the structures of molecules that are difficult to determine with more traditional analytical methods such as NMR or mass spectrometry[35–37], as well as to identify intermediates and products of reactions that have taken place on surfaces[38–41]. In particular, NC-AFM and HR-STM have been employed for examining the internal structure of graphene nanostructures, as the number and type of molecular rings can easily be resolved[42–45]. There are also various examples of the use of these techniques for studying the supramolecular structure of 2D self-assembled layers[21,46–49], although in most of these cases the chemical structure of the molecular components led to an unambiguous assignment of the type of intermolecular bonding.

Here we identify a case—the self-assembly of 3,9-dibromo-*peri*-xanthenoxanthene (3,9-Br$_2$PXX) molecules on a Au(111) surface—where the potential coexistence of different intermolecular interactions presents a formidable challenge for standard forms of SPM. In particular, while conventional STM measurements cannot determine whether 3,9-Br$_2$PXX assembles via HB or XB, only by means of HR-STM are we able to identify XB interactions as the driving force leading to the formation of supramolecular assemblies of 3,9-Br$_2$PXX on Au(111).

## Results

**Molecular assembly.** 3,9-Br$_2$PXX (Fig. 1a) is a dibrominated derivative of PXX, an electron donor molecule[50–53], derivatives of which have previously been employed as *p*-type semiconductors in organic thin-film transistors due to their efficient carrier injection properties, high mobility, and thermal stability[54,55]. The Br and O atoms in 3,9-Br$_2$PXX offer the possibility of both HB and XB intermolecular interactions when arranged into supramolecular arrays on a surface. HB can originate from non-classical weak C–H···O interactions, while the possibility of XB stems from the emergence of the so-called σ-hole[56] on the Br atom (Fig. 1b). Density functional theory (DFT) calculations

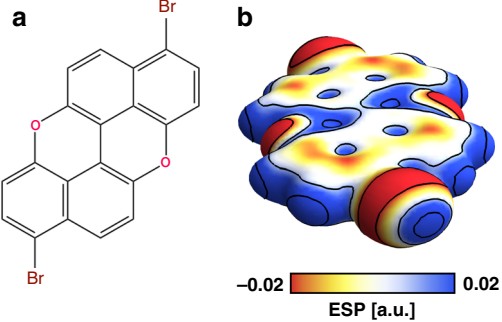

**Fig. 1 Structure and electrostatic potential of 3,9-Br$_2$PXX. a** Molecular structure of 3,9-Br$_2$PXX. **b** Map of the electrostatic potential (ESP) projected on an isosurface (0.001 a.u.) of the electron density, clearly showing the σ-hole and the corresponding electron-rich belt on the Br atoms. Continuous lines separate ESP regions differing by more than 0.01 a.u.

(see Methods) show that there is a region of negative electrostatic potential that forms a belt around the C—Br bonds, while a region of positive potential (the σ-hole) develops in the elongation of the same bond, thus providing the opportunity for Br and O atoms to act as halogen bond donors and acceptors, respectively.

At low coverages, the majority of the 3,9-Br$_2$PXX molecules self-assemble into kagome-type structures (phase 1) that develop in the face-centred cubic (fcc) regions and elbow sites of the Au (111) herringbone reconstruction (Fig. 2a). At 77 K, the peripheries of the kagome islands are often seen to vary between scans, indicating a significant level of molecular mobility. Combined with the lack of any noticeable herringbone distortion, we attribute this to the 3,9-Br$_2$PXX molecules being weakly bound to the Au(111) surface (typically, strongly bound adsorbates result in a lifting or a significant distortion of the herringbone reconstruction[57–60]). Two mirror symmetric orientations of the kagome structures can be found, suggesting that the pro-chiral 3,9-Br$_2$PXX molecules segregate into chiral domains upon adsorption. The kagome structure itself can be thought of as being composed of triangular sub-units that consist of three molecules, each appearing to be bound with the end of one molecule pointing to the side of another (Fig. 2d). While this indicates that the oxygen atoms of 3,9-Br$_2$PXX are involved in intermolecular bonding, the nature of the interaction is not clear, as is discussed below. The triangular sub-units are packed into a hexagonal unit cell with lattice vectors $\mathbf{a} = \mathbf{b} = 2.2 \pm 0.1$ nm, and an angle of $60 \pm 2°$. Full unit cells are rarely found due to the confinement of the assembly in the fcc regions of the herringbone reconstruction.

Coexisting with the kagome structure, a second minority assembly (phase 2, Fig. 2b) is observed at low molecular coverage. Phase 2 consists of irregular structures that, similar to the kagome assembly, are limited to the fcc regions of the Au(111) herringbone reconstruction. Within these irregular islands, variations in molecular shape are observed, with some of the ends of the molecules appearing differently, and some molecules having an extra feature at the sides (Fig. 2e). Increasing the coverage results in the development of a further, denser assembly (phase 3, Fig. 2c), where the molecules are arranged into parallel rows (Fig. 2f). This structure forms extended and compact islands that span both fcc and bcc regions of the substrate and coexist with few remaining phase 1 and phase 2 regions (Supplementary Fig. 1).

Initial modelling of the kagome assembly by overlaying molecular structures onto the STM images results in two distinct possibilities for the type of intermolecular interaction that governs the assembly. The unit cell dimensions and possible

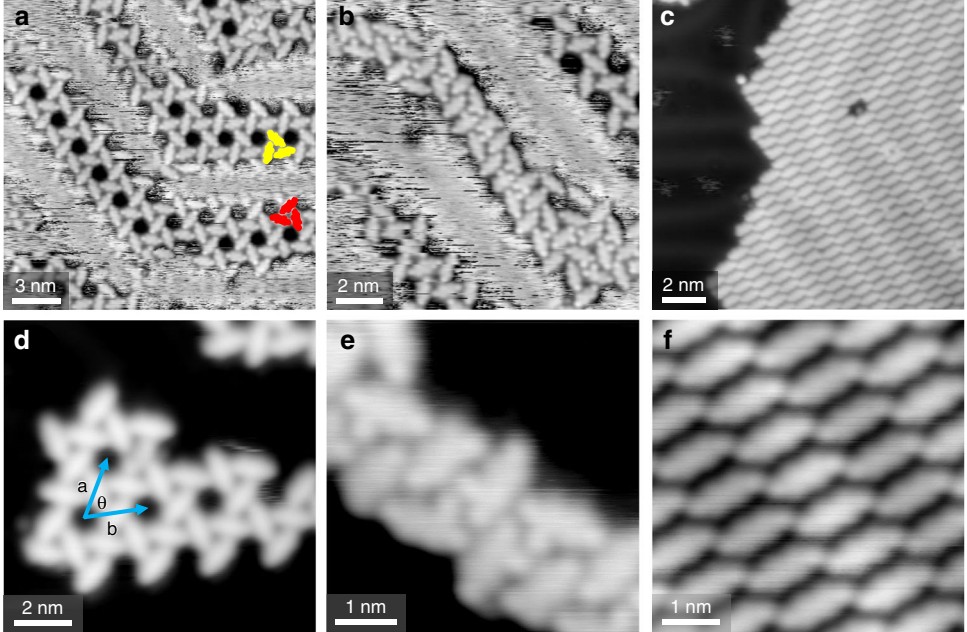

**Fig. 2 STM images of self-assembled structures observed by depositing 3,9-Br₂PXX onto a Au(111) surface. a, d** Majority kagome structure (phase 1) formed at low molecular coverage. Domains of opposite chirality are indicated by red/yellow highlights in **a** and the parameters of the surface unit cell are shown in **d**. **b, e** Minority irregular assembly (phase 2) coexisting with the kagome structure at low molecular coverage. **c, f** Compact assembly (phase 3) that develops at higher molecular coverage. STM images **a** and **b** were acquired at 77 K, **c–f** at 7 K.

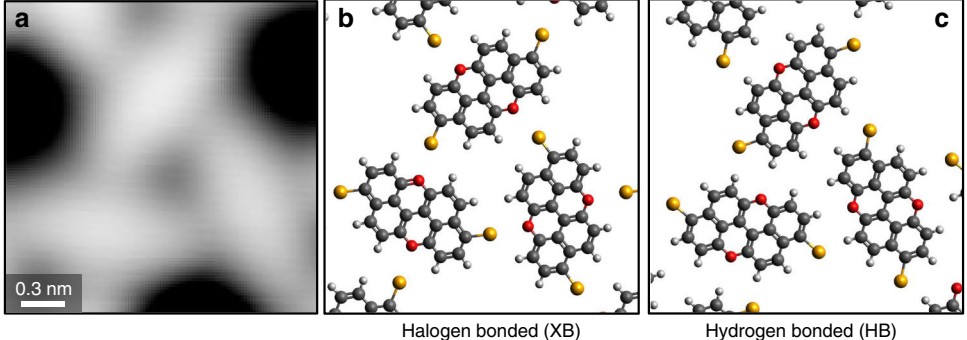

Halogen bonded (XB)　　　　　Hydrogen bonded (HB)

**Fig. 3 Possible intermolecular bonding motifs stabilizing the kagome structure. a** High magnification STM image of the kagome structure highlighting one of its composing triangular sub-units. **b** Possible halogen bonded and **c** hydrogen-bonded structures for the kagome packing.

orientations of the molecule are compatible with either halogen bonding between the Br end groups and the O atoms of adjacent molecules (Fig. 3b), or a non-classical O···H–C hydrogen bond and, possibly, a secondary electrostatic Br···H interaction (Fig. 3c). The relatively featureless appearance of the molecules in typical STM tunnelling conditions does not present an obvious solution to this, even when varying the bias voltage. In this case, the position of the molecular groups (and thus the type of assembly) can therefore not be directly inferred by using standard STM techniques.

**High-resolution STM imaging.** In order to clarify the structure of the kagome assembly, we harnessed the capabilities of HR-STM, which we performed at 7 K using a CO-functionalised tip. In agreement with previous HR-STM experiments and theoretical studies[42,61], intramolecular features could be resolved when approaching the CO tip close to adsorbed molecules (Fig. 4). The rings of the molecule (in particular, the naphthalene aromatic sections) are clearly resolved, as are the positions of the C–Br end groups. This allows us to overlay a molecular model of 3,9-

Br₂PXX in a single orientation for all molecules in the kagome structures of a given chirality (the flipped orientation of 3,9-Br₂PXX matches the HR-STM images of molecules in the kagome structures of the opposite chirality). By doing so, it appears evident that in the triangular sub-units of the kagome packing the Br atoms of one molecule directly face the O atoms of a neighbouring one, as would be expected in XB (Figs. 4d and 3b). Moreover, overlaying scaled molecular models gives an estimated O···Br distance of 3.1 ± 0.1 Å and a C–O···Br angle of 176 ± 2°. The former is smaller than the sum of the van der Waals radii of the two elements (3.37 Å)[62] while the latter is very close to 180°, thus satisfying two established criteria[63] for identifying halogen bonds. All examples of the kagome packing studied by HR-STM were determined to be governed by halogen-bonding interactions. No examples of hydrogen-bonded assemblies could be found, demonstrating that the 3,9-Br₂PXX molecules on Au(111) form only XB at low coverages.

**DFT calculations and image simulations.** To get further insight into the nature of the intermolecular bonding, XB and

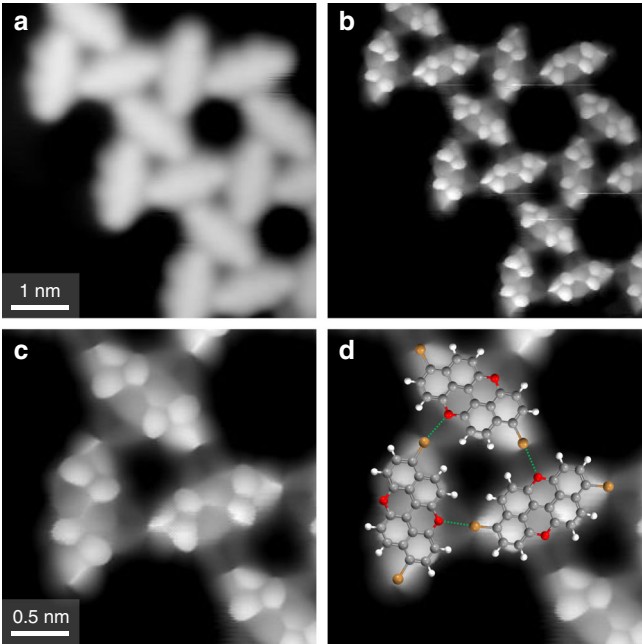

**Fig. 4 High-resolution STM images of the 3,9-Br₂PXX kagome assembly.**
**a** Constant current STM image recorded with typical scanning parameters with a CO tip ($V = 0.51$ V, $I = 160$ pA). **b** Constant height HR-STM image of the same region as **a** ($V = 30$ mV) with the CO tip close to the surface. **c** Smaller scale zoom of the same area seen in b, $V = 30$ mV. **d** Molecular model overlay of **c**, with halogen bonds indicated by green dotted lines. All images were recorded at 7 K.

(hypothetical) HB 3,9-Br₂PXX assemblies were investigated by means of DFT calculations. As the molecules appeared to be weakly adsorbed on the relatively inert Au(111) surface, and since the commensurability of the molecular adlayer could not be determined from the STM measurements, the calculations were performed on free-standing monolayers, taken as reasonable theoretical approximations of the experimental structure. Similar unit cells to those observed experimentally were obtained upon the optimisation of both cell vectors and atomic positions (Supplementary Table 1). Interestingly, the average O⋯H distance obtained for the HB assembly is very similar to the O⋯Br distance expected for the XB assembly, and both are compatible with the experimentally determined value ($3.1 \pm 0.1$ Å). However, the XB appears to be energetically more favourable than the HB assembly by ~80–100 meV/unit cell (the precise value depending on the exchange-correlation functional used, see Supplementary Discussion). This energy difference is indicative that, in agreement with the HR-STM result, the DFT calculations predict that 3,9-Br₂PXX assemblies are held together preferentially by XB interactions as opposed to HB. We argue that the preference for the XB network is due to the strength of the O⋯Br halogen bonds between 3,9-Br₂PXX molecules: evaluating the "binding" energy of a 3,9-Br₂PXX dimer, held together by either XB or HB interactions, results in the former being up to ~25 meV stronger than HB (see Supplementary Discussion).

It is however interesting to compare the results for the XB assembly with those of a hypothetical HB assembly: in Fig. 5 we report the simulated HR-STM images of both assemblies, obtained thanks to the PP-AFM/STM framework of Hapala et al.[64] and Krejčí et al.[61] (see Methods). The agreement between experimental (Fig. 5a) and simulated (Fig. 5b) HR-STM images of the XB assembly is remarkable, further supporting our assignment. On the contrary, the simulated HR-STM image of the (hypothetical) HB assembly (Fig. 5c) not only displays the wrong symmetry with respect to the experimentally observed assembly (Supplementary Fig. 6) but also suggests that the position of the atoms in the HB assembly would produce markedly different features in the HR-STM images. In particular, the bonding region in the XB assembly is characterised by two brighter triangular features crossed by dark streaks (Fig. 5b), while the HB assembly would have an overall darker appearance in (Fig. 5c). We note that the darker streaks observed between the bromine and oxygen atoms of adjacent molecules (Figs. 4c and 5b) should not be interpreted as the 'imaging' of the halogen bonds, as it has been demonstrated that such intermolecular features can result from the relaxation of the flexible CO probe[61,64].

**Analysis of the electron density topology**. The DFT calculations allows us also to investigate the electron density topology which, in conjunction with the structural features discussed earlier, provides an additional probe to determine the presence of an intermolecular XB. To this aim, we have examined the electron density $\rho$ in the proximity of the O and Br atoms, which is plotted in Fig. 6a as a contour map projected onto the molecular plane. It can be clearly recognised that a gradient path connects O and Br and that a saddle point, corresponding to the bond critical point (BCP), is located in between O and Br. This is another determining feature listed in the IUPAC definition of halogen bond[63]. Moreover, Fig. 6b shows the computed bonding charge density difference, i.e. the electron density difference $\Delta \rho = \rho_d - \rho_{m1} - \rho_{m2}$, where $\rho_d$ and $\rho_{m1,2}$ refer to the electron density of the dimer and the monomers, respectively. One can notice the emergence of some charge transfer from the Br atom to the O atom, with a net accumulation of electron density (red regions) along the direction of the O⋯Br bond path. Also this is in agreement with the fact that "the forces involved in the formation of the halogen bond are primarily electrostatic, but polarisation, charge transfer, and dispersion contributions all play an important role"[63]. We finally note that the charge density redistribution is not limited to the close proximity of the O and Br atoms but is characterised by a rather complex pattern as could have been expected from the highly non-uniform molecular electrostatic potential in Fig. 1b. This results in the formation of a sort of binding pocket rather than a single point contact.

**Identification of synthetic impurities**. HR-STM also helps to elucidate the nature of the minority phase 2 by revealing the internal structure of the molecules with an "unusual" appearance and demonstrating that these differ in their configuration of C–Br bonds from the expected 3,9-Br₂PXX structure. In particular, Supplementary Fig. 2 shows that some of these molecules have one terminal C–Br group shifted by one position (i.e., they are 2,9- and 2,8-dibromo isomers), while others are instead tri-brominated, with an extra Br atom covalently bound to a carbon on the side of the molecule (1,3,9-Br₃PXX). The intermolecular interactions controlling the assembly of phase 2 are mostly XB, with some possible contribution from non-classical O⋯H–C hydrogen bonds and secondary electrostatic Br⋯H interactions (Supplementary Fig. 2). Although all the analytical spectroscopic characterisations suggested that the 3,9-Br₂PXX obtained was pure (Supplementary Figs. 11–14), small traces of dibromo isomers and tribromo derivatives might still have been present in the sample due to their low solubility (see experimental procedures in the Supplementary Methods). The fact that these impurities appear in our STM study at a relatively high concentration may be due to a higher rate of sublimation than 3,9-Br₂PXX, resulting in an overrepresentation when deposited onto the surface. In any case, the ability of clearly identifying these molecules by

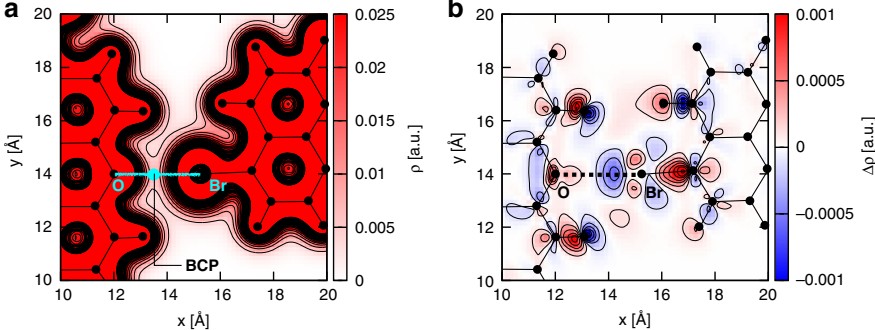

**Fig. 5 Comparison between experimental and simulated HR-STM images of the 3,9-Br₂PXX kagome assembly. a** Constant height HR-STM image (same region as Fig. 4c). **b** Simulated HR-STM image (see text) of the XB assembly. **c** Simulated HR-STM image of the HB assembly. The orange circles highlight the O···Br/H region.

**Fig. 6 Analysis of the electron density topology. a** Contour map of the electron density $\rho$ projected onto the molecular plane $xy$. The location of the O and Br atoms is explicitly shown, together with that of the bond critical point, BCP. **b** Electron density difference $\Delta\rho$, calculated as the difference between the electron density of the dimer and that of the two monomers.

HR-STM when other, conventional spectroscopic characterisation techniques fail to do so, is a further demonstration of the potential of sub-molecular resolution scanning probe microscopy as an analytical tool for chemical structure determination[35]. Interestingly, the minority phase 2 appears to be made exclusively of defective molecules (of various types), while the regular 3,9-Br₂PXX molecules all assemble into the kagome phase 1 structures.

Finally, HR-STM also shows that the compact phase 3 that develops at a higher molecular coverage is kept together by synergistic HB and XB (Supplementary Fig. 3). The molecules are arranged into parallel rows and interact with each other via non-classical O···H–C hydrogen bonds as well as type I halogen bonds between terminal C–Br groups. Interestingly, this assembly is identical to that characterising the 3D crystalline phase of 3,9-Br₂PXX as determined by X-ray diffraction (Supplementary Fig. 4), with one of the lattice parameters being slightly larger, most probably due to the molecule-surface interaction. Phase 3 is almost exclusively composed of regular 3,9-Br₂PXX molecules, although occasional defective molecules (both 2,9-Br₂PXX and 2,8-Br₂PXX dibromo isomers and 1,3,9-Br₃PXX tribromo derivatives) are observed within the compact islands, and partially de-brominated molecules are sometimes found at the edges (Supplementary Fig. 5).

In conclusion, high-resolution STM performed with a CO-functionalised tip is found to be necessary to conclusively identify the arrangement of dibrominated sp²-hybridised molecules within an on-surface self-assembled molecular network. The experimental information provided by HR-STM provides the basis for a theoretical structural, electrostatic, and electron density topology analysis, allowing us to identify with great accuracy the emergence of halogen bonding as the intermolecular interaction that stabilises the observed molecular structures. Whereas standard metallic-tip STM is unable to differentiate between halogen and hydrogen assembly motifs, HR-STM clearly identifies Br···O halogen bonding as the only source of the observed kagome assemblies of 3,9-Br₂PXX on Au(111). DFT calculations support this result by predicting a higher stability of halogen versus hydrogen intermolecular binding while simulations of the HR-STM images demonstrate a remarkable agreement with the experimental data only for the halogen bonding interaction. HR-STM also identified the presence of defective molecules that escaped the scrutiny of traditional analytical methods and thereby fully solved the experimentally observed 2D self-assembly. Building upon the cases in which HR-SPM has been shown to be an invaluable tool for identifying the chemical structure of unknown molecules[35–37] or for determining the intermediates and products of on-surface reactions[38–40], this work exemplifies the unique analytical insight that HR-SPM can give in establishing the nanoscale arrangement of molecules within a supramolecular structure and the nature of the intermolecular interactions ruling the self-assembly. We believe that this methodology can be effectively used when investigating 2D networks originating from the self-assembly of planar molecules on surfaces and that a significant fraction of difficult or controversial molecular structures that have been discussed in the literature over the last decades could be quickly and clearly solved by using this approach.

## Methods

**Scanning tunnelling microscopy measurements**. STM experiments were performed on a low temperature STM under ultra-high vacuum (UHV) conditions. The Au(111) single crystal was cleaned with multiple cycles of Ar⁺ sputtering and annealing. 3,9-Br₂PXX was deposited via sublimation (483 K) onto the Au(111) crystal, held at room temperature. The crystal was then cooled to 77 K or 7 K for

**STM analysis.** Standard STM measurements were carried out with bias voltages ($V$, applied to the sample) in the range of ±2.0 V and tunnelling currents of 50–200 pA. In order to perform high-resolution STM experiments, CO was leaked into the UHV system, adsorbed onto the Au(111) crystal held at ~10 K, and picked up by the STM tip. High-resolution STM images were taken in constant height mode (current channel). STM image analysis was performed with WSxM[65], Gwyddion[66] and LMAPper[67].

**Density Functional theory calculations.** DFT calculations were performed using the mixed Gaussian and Plane-Waves (GPW) method implemented in the CP2K package[68]. As the description of both halogen bonding (XB) and hydrogen bonding (HB) interactions is known to be quite sensitive to the choice of the exchange-correlation (XC) functional[56], we have used two different fully self-consistent non-local XC functionals, namely vdW-DF[69] and optB88-vdW[70], to assess the reliability of our results. Goedecker-type pseudopotentials[71] with four, one, six and seven valence electrons for C, H, O and Br respectively have been employed. The Kohn-Sham orbitals were expanded in a Double-Zeta Valence plus Polarisation (DZVP) Gaussian-type basis set, while the plane wave cutoff for the finest level of the multi-grid[68] has been set to 400 Ry to efficiently solve the Poisson equation within periodic boundary conditions using the Quickstep scheme[68]. Brillouin zone integration was restricted to the supercell Γ-point. We have found that considering a single unit cell (in-plane dimensions of ~20 Å and containing three 3,9-Br$_2$PXX molecules, thus totalling 96 atoms), together with a vacuum region of ~20 Å inserted between the periodic replica of the 2D self-assemblies (along the direction normal to the assemblies planes) is sufficient to ensure an accuracy of the resulting total energy of 3 meV/atom.

**Image simulations.** The simulated HR-STM images have been obtained thanks to the PP-AFM/STM framework of Hapala et al.[64] and Krejčí et al.[61]. The CO-functionalised tip employed experimentally is approximated by a probe particle bonded to the STM tip: this bond is 4 Å long and characterised by a lateral stiffness of 0.5 N/m. The charge of the probe particle is set equal to zero. As the experimental HR-STM images were obtained in constant height using very low bias voltages, all HR-STM simulated images were calculated as constant height dI/dV maps at the energy of the 3,9-Br$_2$PXX highest occupied molecular orbital (HOMO). In order to describe the tunnelling process, we have considered the $s$ and $p$ orbitals of the sample and the $p_x$ and $p_y$ orbitals of the functionalised tip. The Lennard-Jones and electrostatic fields characterising the sample have been obtained using simple point-charge electrostatics[64] and the equilibrium configuration of the system was used, as obtained via the DFT calculations described above. The electronic density of states and the molecular orbitals of the sample have also been obtained via the CP2K code.

Further details on the STM methods as well as the full synthetic details are given in the Supplementary Methods.

## Data availability

All data needed to evaluate the conclusions in the paper are present in the paper and/or the Supplementary Information. Crystallographic data (excluding structure factors) for compound 3,9-Br$_2$PXX reported in this paper have been deposited at the Cambridge Crystallographic Data Centre, under deposition number 1901257. These data can be obtained free of charge from The Cambridge Crystallographic Data Centre via www.ccdc.cam.ac.uk/data_request/cif. Additional data supporting the findings of this study are available from the corresponding author upon reasonable request.

## Code availability

The electronic structure calculations presented in this work have been obtained by the CP2K (version 4.1) package, which is an open source code freely available at https://www.cp2k.org. The simulated HR-STM images have been computed via the PPSTM framework of Ondřej Krejčí, which is freely available via GitHub at https://github.com/ondrejkrejci/PPSTM.

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

## Acknowledgements

G.C. acknowledges financial support from the University of Warwick and from the EU through the ERC Grant "VISUAL-MS" (Project ID: 308115). D.B. gratefully acknowledges the EU through the MSCA-RISE project "INFUSION" (Project ID: 734834), the ERC Grant "COLORLANDS" (Project ID: 280183), and Cardiff University. We thank Dr. Benson Kariuki, Deborah Romito and Nicolas Biot for the X-ray analysis, Dr. Robert L. Jenkins for the help with NMR experiments, Thomas Williams for his support with MALDI-HRMS analysis, and the Analytical Service at the School of Chemistry, Cardiff University. G.C.S. is grateful to the Centre for Scientific Computing at the University of Warwick for providing computational resources. G.C.S. also acknowledges the use of Athena at HPC Midlands+ (funded by the EPSRC on grant EP/P020232/) in this research, as part of the HPC Midlands+ consortium.

## Author contributions

J.L. performed the surface deposition experiments, the STM and HR-STM experiments, analysed the experimental data, and wrote the first draft of the manuscript. H.P. performed DFT calculations. G.C.S. performed the electronic structure calculations, computed the simulated HR-STM images and analysed the results. L.D. and D.B. designed, synthesised, and characterised the molecular module. G.C. conceived and coordinated the research project. All authors participated in discussing the data and editing the manuscript.

## Competing interests

The authors declare no competing interests.
