## [Peer Review File · Nature Communications]

Reviewers' comments:

Reviewer #1 (Remarks to the Author):

The paper uses a combined experimental and theoretical strategy to identify the nature of bonding in self-assembled brominated polycyclic aromatic molecules on Au(111). Thanks to a CO-functionalized tip, high-resolution STM images can be obtained. A recent simulation scheme called probe-particle model allows to simulate such images and relate them to structure.

DFT simulations and experimental/theoretical HR-STM images allow to identify the relative position as compatible with an halogen bonding, thus discarding alternative assemblies involving hydrogen bonding.

The results are convincing and the paper will influence future works in the field concerning approach and methodology. The paper is well written and should be published in Nat. Comm.

How general the presented results are concerning halogen bonding is however not discussed enough in the paper and should be explicitly addressed.

Another point concerns the relatively limited exploiting of DFT results: much more information can be obtained from them and should also be the case here.

Thus, I recommend publication provided the following (minor) additions are introduced:

1) A discussion about the limits of the analysis to the specific systems; how portable is the "footprint" criterion to other superstructures; which caveats must be kept into account when applying the same strategy to a completely different system with putative X-bonding.

2) The authors cite (ref. 63) a detailed "IUPAC definition of halogen bond" which is available online (see <https://www.degruyter.com/downloadpdf/j/pac.2013.85.issue-8/pac-rec-12-05-10/pac-rec-12-05-10.pdf>)

The "distance" criterion between oxygen and halogen is (as expected) just one of the conditions. There are also electronic structure and density-based conditions:

"The forces involved in the formation of the halogen bond are primarily electrostatic, but polarization, charge transfer, and dispersion contributions all play an important role. The relative roles of the different forces may vary from one case to the other."

The author will realize "bonding charge density difference" (A+B density, A density in the geometry of A+B, B density in A+B geometry, difference of the three cases) and so identify which kind of effects are appearing here.

"The analysis of the electron density topology usually shows a bond path (a "bond path" and a "bond critical point" are defined as "Within the topological electron distribution theory, the line resulting from the addition of two gradient paths of the electron density function emanating from the bond critical point located between each two neighbouring atomic basins" and "Within the topological electron distribution theory, a (3, -1) critical point (the point of the gradient field of the electron density within a given neutral configuration in which $\nabla\rho(r,q) = 0$) which is a local maximum in two directions and is a local minimum in the third, i.e. a saddle point in the three directions", see pp. 1928 and 1927, respectively, in ref. [2]) connecting X and Y and a bond critical point (see pp. 1928 and 1927, respectively, in ref. [2]) between X and Y."

Through a Bader analysis the authors will show that this is the case here.

3) Basis set superposition error can affect the binding energies computed with localized basis sets. The authors should discuss and apply the standard Boys and Bernardi counterpoise corrections.

Reviewer #2 (Remarks to the Author):

This paper reports some extremely high quality scanning probe measurements on a novel aromatic compound that exists in a number of polymorphic types on a gold surface.

The main focus of the paper is on the type of bonding between the molecules, where the close proximity of bromine and oxygen atoms is proved by the high resolution experiments.

The paper should definitely be published after the following points have been considered.

The electrostatic potential map in Figure 1 is really nice, but it is not really used to full effect. When the structure is supramolecular structure is drawn, a line between oxygen and bromine is indicated, but this does surely not tell the whole story. Looking at the potential map, there are complementary areas of charge between the oxygen and the "end" of the bromine atom AND between the "sides" of the bromine atom and the hydrogen atoms that form a kind of a pocket on the "side" of the molecule. Indeed, it is interesting that in the HR STM images there is tunnelling observed clearly in this region. Given the size of the bromine atom, I think the argument should be changed a little to take this idea of a binding pocket rather than a single point contact that is a little simplistic perhaps.

Chirality is mentioned, and the chirality of the molecules is observed clearly in the images. It is not clear to this referee if the kind of chiral orbital arrangement in combination with the surface atoms inferred by others is seen here. Perhaps the authors could consider commenting.

Reviewer #3 (Remarks to the Author):

This is an important and well-written paper. The authors have carried out a convincing and compelling joint experimental and simulation study of high resolution STM imaging of a 2D assembly of a brominated polycyclic aromatic molecule, 3,9-Br₂PXX, on Au(111). They achieve high submolecular resolution images by operating in the low bias voltage regime in a mode closely related to that pioneered by Temirov et al. in 2008 [Ref. 28 of the manuscript] and studied more recently by Krejci, Hapala et al. [Ref. 61 of the manuscript.]

The data are of high quality, and the analysis is systematic and convincing. I recommend publication as is, with only one very small change to the abstract. I strongly suggest that the authors remove the word "unambiguously" (from the abstract and elsewhere in the paper). Their work certainly provides very compelling evidence for halogen bonding but there is no direct measurement of that bonding interaction, or direct visualisation of any charge density due to that bond.

The authors may feel that I am "splitting hairs" with this recommendation but there have been a number of somewhat controversial claims about visualising intermolecular interactions in scanning probe images. In their manuscript they are very careful to point out the difficulties with image interpretation of this type. The bond lengths and image contrast seen in the STM images are certainly very suggestive of halogen bonding, for the reasons the authors explain, but are still not direct, entirely incontrovertible evidence.

Response to Referee Report

We thank the Reviewers for having evaluated our manuscript and for their highly appreciative comments and their very useful suggestions. All the issues raised by the Reviewers have now been addressed, leading to an improved revised version of the manuscript.

A point-by-point reply to the Reviewers' comments, together with the description of the corresponding changes made in the article, is reported below.

In the process of answering to the Reviewers' comments we have accumulated additional results which have now been included as a new paragraph and a new figure in the revised version of the manuscript, together with a few other minor changes in both the main paper and the Supplementary Information (SI). Changes made to the revised versions of manuscript have been indicated by blue text.

Reviewer #1

I recommend publication provided the following (minor) additions are introduced: 1) A discussion about the limits of the analysis to the specific systems; how portable is the "footprint" criterion to other superstructures; which caveats must be kept into account when applying the same strategy to a completely different system with putative X-bonding.

We thank the Reviewer for this comment, that prompted us to reflect more extensively on the range of applicability of our analysis. As showcased in this work, the combination of HR-SPM experiments and DFT computer simulations that we have employed constitutes a very robust analytical framework. We believe this hybrid methodology can be effectively used when investigating 2D networks originating from the self-assembly of sp^2 -hybridized molecules on metallic surfaces. Arrays of planar molecular structures such as those of polycyclic aromatic hydrocarbon are thus ideal candidates to be investigated with this combined method. On the other hand, assemblies formed by non-planar molecules (e.g. by molecules with several sp^3 -hybridized carbons) or intrinsically non planar assemblies (e.g. different types of metal-organic networks) might pose significant experimental challenges in terms of the HR-SPM measurements. At the same time, molecular species characterised by a strong propensity for charge transfer might benefit from the usage of wavefunction methods in addition to DFT calculations. However, we expect that in the vast majority of 2D self-assembled systems the topological analysis of the electron density – which indeed has been suggested by the Reviewer themselves (see below) – together with the structural information from HR-SPM, provide a comprehensive set of data that allows to pinpoint with great accuracy the emergence of either hydrogen- or halogen-bonding. To clarify this point we have added the following sentence in the revised manuscript (on Page 10): “We believe that this methodology can be effectively used when investigating 2D networks originating from the self-assembly of planar molecules on surfaces”.

2) The authors cite (ref. 63) a detailed "IUPAC definition of halogen bond" which is available online. The "distance" criterion between oxygen and halogen is (as expected) just one of the conditions. There are also electronic structure and density-based conditions: "The forces involved in the formation of the halogen bond are primarily electrostatic, but polarization, charge transfer, and dispersion contributions all play an important role. The relative roles of the different forces may vary from one case to the other."

The author will realize "bonding charge density difference" (A+B density, A density in the geometry of A+B, B density in A+B geometry, difference of the three cases) and so identify which kind of effects are appearing here.

"The analysis of the electron density topology usually shows a bond path (a "bond path" and a "bond critical point" are defined as "Within the topological electron distribution theory, the line resulting from the addition of two gradient paths of the electron density function emanating from the bond critical point located between each two neighbouring atomic basins" and "Within the topological electron distribution theory, a (3, -1) critical point (the point of the gradient field of the electron density within a given neutral configuration in which $\nabla\rho(r,q) = 0$) which is a local maximum in two directions and is a local minimum in the third, i.e. a saddle point in the three directions", see pp. 1928 and 1927, respectively, in ref. [2]) connecting X and Y and a bond critical point (see pp. 1928 and 1927, respectively, in ref. [2]) between X and Y." Through a Bader analysis the authors will show that this is the case here.

We are very grateful to the Reviewer for pointing out the relevance of the electron density-based criterion for the identification of halogen bonds. In particular, we highly appreciate their detailed suggestions on how to further strengthen our analysis of the halogen bonds in the assemblies we have investigated.

Following the advice of the Reviewer, we have investigated the electron density ρ in the proximity of the O...Br halogen bond and computed the bonding charge density difference. The results are summarised in the figure below, which has been added to the main manuscript as the new Figure 6. In panel (a) we report the contour map (projected onto the molecular plane xy) of the electron density in the proximity of the O and Br atoms. **As predicted by the Reviewer, we found that a gradient path connects O and Br and that a saddle point, corresponding to the bond critical point (BCP), is located in between O and Br.** In panel (b) we show the electron density difference $\Delta\rho = \rho_d - \rho_{m1} - \rho_{m2}$, where ρ_d and $\rho_{m1,2}$ refer to the electron density of the dimer and the monomers, respectively: one can notice the emergence of some charge transfer from the Br atom to the O atom, with an net accumulation of electron density (red regions) along the direction of the O...Br bond path. This is in accordance with the IUPAC description of the forces involved in the formation of the halogen bond, also cited by the Reviewer.

Following the Reviewer's suggestion, we have added a new figure (Fig. 6) and a new paragraph to the main paper which contains the discussion above, based on the analysis of the electron density (pages 7 and 8). Moreover, on page 6, we have included the experimental evaluation of the C-O...Br angle which, being equal to $176 \pm 2^\circ$, is extremely close to 180° , and thus represents a further criterion for identifying halogen bonds established in the IUPAC definition. Finally, we have also added a sentence in the paper abstract, and one in the conclusion, highlighting the results of this analysis.

3) *Basis set superposition error can affect the binding energies computed with localized basis sets. The authors should discuss and apply the standard Boys and Bernardi counterpoise corrections.*

The Boys-Bernardi counterpoise correction (CPC) the Reviewer refers to [S. F. Boys and F. Bernardi, *Mol. Phys.* 19, 553 (1970)] seeks to eliminate the so-called "basis set superposition errors" (BSSEs). These arise from the fact that, when dealing with first-principle calculations of weakly bound species (particularly dimers such as the ones investigated here) using localised basis sets, the dimer may be artificially stabilised, as one of the two monomers leverages the extra basis functions from the second monomer to describe its electron distribution - and vice versa.

We note that the effectiveness of computing the CPC in the case under investigation might be somewhat limited for a number of reasons:

- i. In our case – where we are using DFT as opposed to wavefunction-based methods – the choice of the exchange-correlation functional as well as of the basis sets (which we have both thoroughly validated, see the SI) are a much more significant source of uncertainty. This is supported by a recent publication [Mentel, Ł.M., and Baerends, E.J., *Can the Counterpoise Correction for Basis Set Superposition Effect Be Justified?* *Journal of Chemical Theory and Computation* 10, 252–267 (2014)], stating that *"it has also often been observed that CPC does not always improve the interaction energy compared to a known benchmark result or a very good estimate based for instance on extrapolation to the complete basis set"*.
- ii. The fragmentation of the systems needed to apply the CPC is not straightforward for systems involving substantial intramolecular interactions (such as the assembly we are investigating in this work).
- iii. We have verified that even the relatively small basis sets we are using (DZVP) are enough to converge energies and geometries within a few meV and 0.01 Å, respectively.

Nevertheless, we have followed the Reviewer's advice and proceeded to calculate the CPC in the case of both halogen- and hydrogen-bonded dimers: while the extent of the CPC is not negligible (~10 meV), it is important to note that it only leads to a systematic shift in the binding energies which is smaller than the difference we have found when using different exchange-correlation functional and, crucially, does not lead to any significant structural change.

Reviewer #2

The electrostatic potential map in Figure 1 is really nice, but it is not really used to full effect. When the structure is supramolecular structure is drawn, a line between oxygen and bromine is indicated, but this does surely not tell the whole story. Looking at the potential map, there are complementary areas of charge between the oxygen and the "end" of the bromine atom AND between the "sides" of the bromine atom and the hydrogen atoms that form a kind of a pocket on the "side" of the molecule. Indeed, it is interesting that in the HR STM images there is tunnelling observed clearly in this region. Given the size of the bromine atom, I think the argument should be changed a little to take this idea of a binding pocket rather than a single point contact that is a little simplistic perhaps.

We thank the Reviewer for remarking the need to improve the discussion about the details of the electrostatic interaction (and thus of the nature of the bonding) between the bromine

and the oxygen atom. We think that the thorough analysis of the charge density distribution that has now been added to the main paper as Fig. 6 and the corresponding new paragraph (please see also response to comment 2 of Reviewer 1) fulfils this requirement. In particular, prompted by the Reviewer's comment, we have now added two sentences on page 8 of the main paper, to highlight the relationship between the non-trivial electrostatic potential map in Fig. 1b and the electron density distribution of the assembly: "*We finally note that charge density redistribution is not limited to the close proximity of the O and Br atoms but is characterised by rather complex pattern as could have been expected from the highly non-uniform molecular electrostatic potential map in Fig. 1b. This results in the formation of a sort of binding pocket rather than a single point contact.*"

Chirality is mentioned, and the chirality of the molecules is observed clearly in the images. It is not clear to this referee if the kind of chiral orbital arrangement in combination with the surface atoms inferred by others is seen here. Perhaps the authors could consider commenting.

The 3,9-Br₂PXX molecule is prochiral and can be converted to a chiral structure upon surface deposition: when a prochiral molecule is absorbed on a substrate, only symmetry elements perpendicular to the surface plane are allowed, reducing the dimensionality of the molecule and breaking its symmetry. This phenomenon has been very well documented in the literature and is also observed in our study, where the deposition of 3,9-Br₂PXX on the Au(111) surface creates two types of chiral molecular adsorbates, mirror images of each other, both present as homochiral segregated phases. Establishing whether the chiral supramolecular structures observed in our STM images are also the expression of chirality at further levels would go far beyond the scope of this work and has therefore not been additionally investigated. We would however be tempted to exclude that, in our case, a further cause of chirality could derive from a molecular distortion upon adsorption or from the formation of chiral molecular orbitals due to an asymmetric charge transfer between the molecules and the substrate. In fact, the cases reported in the literature where these two effects arise, are mostly characterised by a significant molecule-surface interaction implying a considerable degree of chemical bonding. On the contrary, both experimental and theoretical evidence seems to indicate a rather weak interaction between the 3,9-Br₂PXX molecules and the Au(111) substrate.

Reviewer #3

I recommend publication as is, with only one very small change to the abstract. I strongly suggest that the authors remove the word "unambiguously" (from the abstract and elsewhere in the paper). Their work certainly provides very compelling evidence for halogen bonding but there is no direct measurement of that bonding interaction, or direct visualisation of any charge density due to that bond.

The authors may feel that I am "splitting hairs" with this recommendation but there have been a number of somewhat controversial claims about visualising intermolecular interactions in scanning probe images. In their manuscript they are very careful to point out the difficulties with image interpretation of this type. The bond lengths and image contrast seen in the STM images are certainly very suggestive of halogen bonding, for the reasons the authors explain, but are still not direct, entirely incontrovertible evidence.

We thank the Reviewer for their comment and we agree that even our "very compelling evidence" is not a definitive proof for halogen bonding. We feel that our additional analysis

of the electron density topology (please see response to comment 2 of Reviewer 1) further supports this claim but we agree that this is still slightly short of being an incontrovertible (experimental) evidence. We have thus removed the adverb “unambiguously” from the manuscript.

REVIEWERS' COMMENTS:

Reviewer #1 (Remarks to the Author):

The authors have addressed all my comments and suggestions extensively and satisfactorily. They included novel results and made the already excellent scientific frame even more amazing. The paper can be now published as is.

Reviewer #2 (Remarks to the Author):

The authors have responded excellently to all comments by the referees.
The article should be published.